# Effects of Stocking Density on Fatty Acid Metabolism by Skeletal Muscle in Mice

**DOI:** 10.3390/ani12192538

**Published:** 2022-09-22

**Authors:** Qiuyan Chen, Xiaohui Li, Jiarun Cui, Caiyun Xu, Hongfei Wei, Qian Zhao, Hongli Yao, Hailong You, Dawei Zhang, Huimei Yu

**Affiliations:** 1Center of Animal Experiment, College of Basic Medical Sciences, Jilin University, Changchun 130021, China; 2Key Laboratory of Pathobiology, Ministry of Education, Department of Pathophysiology, College of Basic Medical Sciences, Jilin University, Changchun 130021, China

**Keywords:** SPF Kunming mice, intestinal flora, fatty acids, stocking density, muscle quality

## Abstract

**Simple Summary:**

Appropriate stocking density is one of the most basic guarantees for experimental animals, and it is also a prerequisite for ensuring the accuracy and credibility of experimental data science. Stocking density, which is related to gut microbiota, affects laboratory mouse health. The aim of this study is to evaluate whether the stocking density of experimental animals affects the gut microbiota, fatty acid metabolism, and muscle quality, and ultimately affects their welfare. According to our results, medium stocking density improves gut microbiota, fatty acid metabolism, and muscle quality in experimental animals.

**Abstract:**

Specific pathogen-free (SPF) grade laboratory animals are kept in specific cages for life. The limited space could affect the characterization of colonization and dynamic changes related to gut microorganisms, and affect adipokines, even further affecting the fat synthesis and muscle quality of animals. The objective of this study was to analyze the stocking density on the dynamic distribution of gut microbiota, fat synthesis and muscle quality of SPF grade Kunming mice. Three housing densities were accomplished by raising different mice per cage with the same floor size. Kunming mice were reared at low stocking density (LSD, three mice a group), medium stocking density (MSD, 5 mice a group), and high stocking density (HSD, 10 mice a group) for 12 weeks. The results demonstrated that the stocking density affected intestinal microbial flora composition. We found that compared with the MSD group, the abundance of Lactobacillus in the LSD group and the HSD group decreased, but the abundance of unclassified_Porphyromonadaceae increased. Moreover, fat synthesis and muscle quality were linked to the intestinal microbial flora and stocking density. Compared with the LSD group and the HSD group, the MSD group had a more balanced gut flora, higher fat synthesis and higher muscle quality. Overall, this study demonstrated that stocking density could affect gut microbiota composition, and reasonable stocking density could improve fat synthesis and muscle quality. Our study will provide theoretical support for the suitable stocking density of laboratory animals.

## 1. Introduction

SPF mice are the most widely used species of animals in laboratory research, and surprisingly little is known about how the behavioral biology of the mouse relates to social and physical aspects of laboratory housing conditions [1,2]. More attention should be paid to the day-to-day activities of mice in their cages, which could also influence research outcomes [3]. The stocking density of animals may influence the behavioral predictors of welfare [4]. Some studies demonstrated that rodents of a high stocking density exhibited reduced muscle enzyme activities as well as decreased muscle strength on an inclined plane test [5], as well as increased adiposity [6]. Some studies demonstrated that mice of a low stocking density exhibited increased energy metabolism [7]. Thus, optimizing the stocking density of SPF mice can not only effectively save the cost of feeding, but also ensure the accuracy of animal experiments and animal welfare.

The gut microbiota acts as a bridge between host health, nutrition, and survival conditions [8]. The capacity of the gut microbial composition or gene-expression pattern may change in response to the host’s physiological changes, thus contributing to the phenomic plasticity, and the variation of the external environment facilitates host acclimation and adaptation to environmental change through the change of gut microbiota [9]. These aspects imply that the gut microbiome may vary with laboratory animal stocking densities. However, how stocking density affect microorganisms remains unknown.

Intestinal microbiota has an important role in skeletal muscle development and lipid metabolism in animals, and the intake of probiotics/prebiotics affects skeletal muscle development and metabolism [10,11]. In addition, skeletal muscle properties are transmissible via fecal microbiota transplantation (FMT) [12]. Gut microbial composition correlates with skeletal muscle fat deposition [13]. A hypothetical mechanism for the interaction of gut microbiota and host fat metabolism is related to the regulation of lipoprotein lipase (LPL) [14]. LPL is a key enzyme in lipid metabolism. It increases fatty acid uptake in tissues by breaking down very low-density lipoproteins and triglycerides in chylomicrons in blood. When intramuscular fat is synthesized, it controls the uptake of fatty acids by muscle and adipose tissue. Sterol regulatory element binding protein 1 (SREBF1) is an important nuclear transcription factor for fatty acid synthesis, which promotes fat deposition in skeletal muscle by upregulating the expression of fatty acid synthesis key molecules’, fatty acid synthase (FASN) and acetyl-CoA carboxylase (ACACA) [15]. Patatin-like phospholipase domain protein 2 (PNPLA2), a very important intracellular triglyceride hydrolase, is responsible for the breakdown of fat. It can promote the release and oxidation of free fatty acids and improve lipid metabolism in the liver. Another mechanism involves the production of short-chain fatty acids (SCFAs), the main fermentation product of indigestible carbohydrates available to intestinal flora [16]. They are rapidly absorbed and utilized by the host and affect lipid metabolism and adipose tissue at several levels [17]. SCFAs can enter the systemic circulation and be absorbed by skeletal muscle cells to regulate muscle function [18]. LPL and SCFAs are also related to intramuscular fat (IMF) content [19], but their role in intestinal microorganism-mediated intramuscular fat metabolism is not clear [20].

The relationship between stocking density and intestinal flora in vivo was measured by metagenomic sequencing. In this study, we analyzed the effect of stocking density on the diversity of gut microbiota in SPF Kunming mice and explored the role of gut microbiota-mediated intramuscular fat metabolism under the influence of stocking density. Meanwhile, reasonable stocking density plays an important role in ensuring the stability and reliability of experimental results and laboratory animal welfare.

## 2. Materials and Methods

### 2.1. Experiment Animals and Treatment Protocols 

Kunming mice, male, six-week-old, 18–20 g, SPF grade were obtained from Changchun Yisi Experimental Animal Technology (Changchun, China). Kunming mice were kept at a constant temperature (22 °C), with a light/dark cycle of 12 h. Animals had ad libitum access to standard rodent chow and tap water in 530.1 × 10^−5^ m^3^ cages all the time. Cages and water bottles were changed once weekly in the dark phase under red light. Briefly, all mice initially allotted to 1 of 3 stocking density treatments: LSD (176.7 × 10^−5^ m^3^/mouse, 3 mice/cage); MSD (106.02 × 10^−5^ m^3^/mouse, 5 mice/cage); HSD (53.01 × 10^−5^ m^3^/mouse, 10 mice/cage) were followed in further experiment analysis. All animals used in the procedures were handled according to the Guide for the Care and Use of Laboratory Animals, and the in vivo experimental methods were approved by the University Committee in the Use of Animals of Jilin University, China.

### 2.2. Sample Collection

Kunming mice in each group were fasted from food and water for 8 h. After euthanasia by carbon dioxide inhalation, the colorectal was quickly cut off, and the fresh mouse feces were collected and marked in the fecal collection box. After taking the feces, the gastrocnemius muscle was taken, and all the samples were stored in liquid nitrogen for 2 weeks. 

### 2.3. Genomic DNA Extraction and PCR Amplification

Genomic DNA was extracted using the E.Z.N.A Mag-Bind Soil DNA Kit (Omega, Shanghai, China) extraction kit. The DNA concentration was measured with the Qubit 3.0 fluorometer (Thermo Fisher Scientific, Waltham, MA, USA). PCR amplification was performed using Ex Taq DNA polymerase (TaKaRa, Otsu, Japan) and primers 341F (5-CCTACGGGNGGCWGCAG-3) and 805R (5-GACTACHVGGGTATCTAATCC-3). PCR was initiated at 94 °C for 3 min, followed by 5 cycles of 94 °C for 30 s, 45 °C for 20 s, and 65 °C for 30 s and 20 cycles of 94 °C for 20 s, 55 °C for 20 s, and 72 ℃ for 30s, with a final extension step at 72 °C for 5 min. Negative controls were run for each barcoded primer. No PCR product for the negative controls was observed on a 1.5% agarose gel. The analyses of the sequencing data were mainly performed using Miseq (Illumina Inc., SanDiego, CA, USA). We spliced and filtered the original data to filter out contaminated data, such as chimera sequences, nucleotide mismatch, and ambiguous character reads, to obtain accurate and reliable adequate data. The identified 16S rRNA sequences (the effective sequencing tags) were analyzed using Uparse software. Sequences sharing ≥97% identity were considered at the same taxonomic position, which were assigned to one operational taxonomic unit (OTU).

### 2.4. 16S rRNA Sequencing and Bioinformatics Analysis

Miseq sequencing and the sequenced PE reads was first spliced according to the overlap relationship. After distinguishing the samples, the sequence quality was controlled and filtered, and then OTU cluster analysis and species taxonomic analysis were carried out. Based on the results of the OTU cluster analysis, a variety of diversity indices of OTU can be analyzed, as well as the detection of sequencing depth; based on the taxonomic information, the statistical analysis of the community structure can be carried out at various classification levels. To assess whether the 16S rRNA sequences could include all the bacteria in the samples, we conducted the alpha diversity analysis, represented by rarefaction curves. Alpha diversity was analyzed using the Quantitative Insights into Microbial Ecology (QIIME) software package (Version 1.7.0) (Boulder, CO, USA). To evaluate alpha diversity, we conducted the analysis of community richness with the Chao1, Shannon, Simpson and Coverage. Abundance curves ranked based on the OTU level were generated to compare the richness and evenness of the OTU among the samples. 

### 2.5. RT-qPCR

Use Trizol Reagent (Invitrogen) to extract the total RNA from the tissues. Use EasyScript First Strand cDNA Synthesis SuperMix (Sangon, Shanghai, China) to digest and reverse transcribe the genomic DNA according to the manufacturer’s instructions. For qPCR analysis, TranStart Green qPCR SuperMix (Sangon, Shanghai, China) was used to amplify cDNA. The cycle parameters are 94 °C 5 seconds, 50 °C–60 °C 15 seconds and 72 °C 10 seconds for 40 cycles. Then, the dissociation curve was analyzed to check the specificity of PCR. The CT value was measured in the exponential amplification phase. The relative expression level of the target gene (defined as multiple change) was measured by the 2^−ΔCT^ method. Actb was used as an internal reference, standardizing the expression level to the multiple changes detected in the corresponding control cells, which was defined as 1.0. And all the primer sequences are shown in Table 1.

### 2.6. Database 

16s rRNA database of bacteria and archaea:

RDP 16S database: default database, http://rdp.cme.msu.edu/misc/resources.jsp (accessed on 27 October 2021)

Silva 16S database: http://www.arb-silva.de/ (accessed on 27 October 2021)

NCBI 16S database: http://ncbi.nlm.nih.gov/ (accessed on 27 October 2021)

GTDB database: full-length default database https://gtdb.ecogenomic.org/ (accessed on 27 October 2021)

### 2.7. Statistical Analysis

All experimental data represent three independent experiments, and each repeated experiment was divided into the low stocking density group (3 mice/cage), the middle stocking density group (5 mice/cage) and high stocking density group (10 mice/cage). All data are given as mean values ± standard deviation (S.D). Results between the two groups were compared using the Student’s *t* test. * *p* < 0.05 was considered a statistically significant difference. Statistical analysis was performed with GraphPad Prism 8.0 (La Jolla, CA, USA).

## 3. Results

### 3.1. Effects of Stocking Density on Body Weight and Shape of Spf Mice 

As shown in Figure 1, after 12 weeks of feeding, different stocking densities had an effect on the body weight of Kunming mice. The initial body weight of the mice in each group is basically the same; after 12 weeks, it can be observed that the weight of the mice in the LSD group is the highest, and the weight of the mice in the HSD group is the lowest, which demonstrated a negative correlation between feeding density and weight. In addition, the mice in the HSD group demonstrated a decreased amount of drinking. The above results suggested that different densities of feeding had certain effects on the quality of Kunming mice.

### 3.2. The Intestinal Contents of Kunming Mice with Different Feeding Densities Were Analyzed by 16S rRNA

The sample sequences of the LSD group, the MSD group and the HSD group were analyzed by OTU cluster analysis. The microbial community was evaluated by the high-throughput sequencing of 16S rRNA gene. The sequencing results in Table 2 showed that a total of 582,020 sequences were generated from three group samples (LSD = 172,621, MSD = 182,646, HSD = 226,753). After optimizing the preliminary data, a total of 416,751 bacterial sequences were acquired from all the samples, ranging from 120,409 to 169,981 sequences per group. Figure 2 showed that among the total 594 OTUs, there were 21 unique OTUs in the LSD group, 20 unique OTUs in the MSD group, and 54 unique OTUs in the HSD group. There were 383 OTUs in all three groups. Moreover, there were differences in the number of OTU among different groups, and the number of OTU was as follows: MSD > LSD > HSD.

### 3.3. Alpha Diversity Analysis 

The Chao index represent the indices of community distribution abundance, and the larger the value, the higher the community distribution abundance. Figure 3A showed that the Chao Indexes were the highest in the HSD group but was lowest in the LSD group. Shannon index reflects community diversity, and the larger the value, the higher the community diversity. Figure 3B showed the Shannon index was highest in the LSD group but was lowest in the HSD group. Figure 3C showed that the Simpson index was lowest in the LSD group but was highest in the HSD group. Additionally, Good’s coverage estimates were over 99% per sample, suggesting that the evenness and abundance of all samples were satisfactory (Figure 3D). 

The dilution curve is used to verify whether the sample is sufficient. As shown in the dilution curve in Figure 4A, the dilution curve of intestinal flora of mice under various stocking densities tends to be flat, indicating that the number of samples is sufficient. Rank-abundance curve directly reflects the abundance and rarity of OTUs. The results, as shown in Figure 4B, demonstrate that the Rank-abundance curves of Kunming mice in different feeding density groups tended to be flat, and the number of OTU was more than 300. The results demonstrated that the diversity of flora in the above groups was rich and evenly distributed. 

### 3.4. Multi-Stage Species Composition Analysis of Single Sample 

We used partial least squares discriminant analysis (PLS-DA) to study the intestinal flora of mice in LSD, MSD, and HSD groups. PLS-DA method is used to classify the samples of each group well, which shows that our classification method is effective (Figure 5A). We analyzed the relative alterations of preponderant taxa at phylum and genus taxonomical levels in the gut’s bacterial. After the bacterial communities were taxonomically assigned, we found that almost all sequences belonged to the following three phylums, namely Firmicutes, Bacteroidetes, and Proteobacteria. Then, further analysis found that Firmicutes accounted for the highest proportion, followed by Bacteroidetes. The phylum composition of various densities is shown in Table 3 and Figure 5B,D. To further study the differences in the composition of intestinal microflora among different groups, we analyzed differences between groups at the genus level. Results as shown in Figure 5C,E and Table 4; the abundance of Lactobacillus, Akkermansia, Bifidobacterium, Turicibacter, and unclassified_Desulfovibrionaceae in the LSD group was significantly lower than that in the MSD group. However, the abundance of unclassified_Porphyromonadaceae, Alloprevotella, Barnesiella, Bacteroides, and Streptococcus in the LSD group was higher than that in the MSD group. Compared with the MSD group, the genera of unclassified_Porphyromonadaceae, Akkermansia, Streptophyta, unclassified_Bacteria, and Acinetobacter increased significantly, while the number of unclassified_Lachnospiraceae, unclassified_Erysipelotrichaceae, Bifidobacterium, Alloprevotella, and unclassified_Desulfovibrionaceae decreased in the HSD group. In the colorectum of Kunming mice reared at the genus level, Lactobacillus has the highest abundance of all kinds of density mice, and unclassified_Porphyromonadaceae has the second highest abundance. These two genera are the dominant genera.

### 3.5. Effects of Stocking Density on Fat Metabolism of SPF Kunming Mice 

As shown in Figure 6A, the *LPL* expression of SPF Kunming mice in the MSD group was higher than that in the LSD group and the HSD group. The expression of LPL in Kunming mice in the LSD was the lowest, followed by the HSD group. To further detect the effects of different feeding densities on fat synthesis in Kunming mice, it detected the effects of different feeding densities on *FASN* and *ACACA* gene expression. The results are as shown in Figure 6B,C, and the expression levels of *FASN* and *ACACA* genes in the MSD group were significantly higher than those in the LSD density group and the HSD density group, indicating that the fat synthesis ability of mice in the MSD group was significantly enhanced. The results shown in Figure 6D demonstrated that the level of *PNPLA2* in the MSD group was lower than that in the LSD group and the HSD group, indicating that fat decomposition decreased in the MSD group and significantly increased in the HSD group. 

### 3.6. Effects of Stocking Density on Muscle Function of Spf Kunming Mice

The expression levels of *MYH1*, *MYH2,* and *MYH4* genes encoding fast muscle fibers in the MSD group were significantly higher than those in the LSD group and the HSD group showed in Figure 7A–C. However, the expression level of the *MYH7* gene encoding slow muscle fiber in the MSD group was significantly lower than that in the LSD group and the HSD group (Figure 7D). These results suggest that mice fed with the middle stocking density have a stronger ability to synthesize fast muscle fibers, while mice fed with low stocking density and high stocking density have a stronger ability to synthesize slow muscle.

## 4. Discussion

Previous studies demonstrated that the stocking density has been demonstrated to affect the growth performance and health status of animals [21]. Digestive microbiotas are also affected by stocking density [22]. Bacteroides can decompose complex polysaccharides into short-chain fatty acids (SCFA) [23]. SCFA plays a lipid-regulating role by up-regulating the gene of propionic acid fatty acid, inhibiting the activity of hepatic fat synthase and regulating the distribution of cholesterol in the blood and liver, thereby reducing serum triglyceride and cholesterol levels, helping to reduce fat synthesis and promote fat breakdown [24,25]. The microorganisms of Bacteroides improve the digestive efficiency in herbivores and animals [26,27]. Firmicutes is the main decomposer of intestinal saturated fatty acids [28], and its members help to produce enzymes involved in fermenting nutrients and potentially produce vitamin B, and contribute to better absorption of calories from food, eventually leading to obesity [29,30]. In our study, we compared gut microbiota in the colorectum of Kunming mice in the LSD, MSD, and HSD groups. Compared with the MSD group, the Lactobacillus of the LSD group decreased at the genus level, and the unclassified_Porphyromonadaceae increased at the genus level, while the Lactobacillus of the HSD group remained basically unchanged and the unclassified_Porphyromonadaceae increased at the genus level. Lactobacillus belongs to Firmicutes and unclassified_Porphyromonadaceae belongs to Bacteroides. Both play an important role in regulating the synthesis and decomposition of fat. The ratio of Firmicutes/Bacteroidetes(F/B), which is a biomarker of obesity, is related to host energy capture and fat deposition [31]. When the Firmicutes were abundant, the expression of genes controlling fat synthesis increased and the expression of genes controlling fat decomposition decreased [32,33]. In the intestines of Kunming mice rich in Bacteroides, the expression of genes controlling fat decomposition increased and that of genes controlling fat synthesis decreased. Therefore, the fat synthesis ability of MSD group was higher than that of the LSD group and HSD group.

Lipoprotein esterase (LPL) is responsible for fat synthesis [34]. It can decompose triglycerides in blood with very low-density lipoprotein and chylomicron and increase fatty acid uptake in adipose tissue, which is positively correlated with intramuscular fat deposition [35,36]. *FASN* and *ACACA* genes cooperatively mediate the response of hepatocytes to insulin and glucose and regulate lipogenic transcription factors to bind with fatty acid synthase, thus promoting the accumulation of triglycerides in the liver and increasing the synthesis of hepatic fat in the host [37,38]. By regulating the activity of adenosine monophosphate activated protein kinase, these genes affect a series of downstream reactions on the oxidative decomposition of fat, and maintain the process of fat synthesis [39]. In our experimental study, the abundance of Bacteroides was high, but the abundance of Bacteroides was low in the MSD group; thus, the regulation of flora increased the expression of genes promoting fat synthesis and decreased the level of gene expression that promoted fat decomposition.

The composition of skeletal muscle fiber types is related to fat content and muscle distribution, which directly affects muscle quality [40]. The area and density of muscle fiber are closely related to intramuscular fat content [41]. Skeletal muscle has four myosin heavy chain (MyHC) subtypes, namely MYH7 (type I), MYH2 (type IIA), MYH1(type IIX), and MYH4 (type IIB) [42]. Genes such as *MYH7*, *MYH2, MYH1,* and *MYH4* affect the contraction–relaxation activity of skeletal muscle and are involved in determining muscle composition [43]. *MYH1, MYH2,* and *MYH4* encode fast muscle fibers, while *MYH7* encodes slow muscle fibers [44]. Type I muscle fibers encoded by *MYH7* have a small diameter and a large number of muscle fibers per unit area, and therefore, have a high intramuscular fat (IMF) content [45]. There was a negative correlation between MyHC IIb encoded by *MYH4* and IMF [46]. The expression of *MYH7* was low, the expression of *MYH4, MYH2, MYH1* was high, and the content of IMF was low in the LSD group. In the HSD group, the food distribution was not uniform due to the HSD group, and the fast-twitch fibers dominated due to less exercise space. These evidences suggest that the lipid metabolism and muscle fiber development of skeletal muscle are related to the characteristics of intestinal microflora.

## 5. Conclusions

Stocking densities affect the welfare of experimental animals, and thus constitute key concerns. The results of the present study suggest that the percentages of beneficial microorganisms involved in regulating intestine functions, reducing incidences of gut diseases, and driving the evolution of the immune system, were significantly increased, and the percentages of harmful microorganisms were reduced, in the MSD group. The ratio of F/B is related to lipid metabolism, and the mice in the MSD group had increased fat synthesis and decreased fat fraction. Stocking density affected the muscle quality of mice, and our results demonstrated that mice in the MSD group had higher fast muscle synthesis capacity. Consequently, the middle stocking density can improve animal welfare and save the cost of feeding.

## Figures and Tables

**Figure 1 animals-12-02538-f001:**
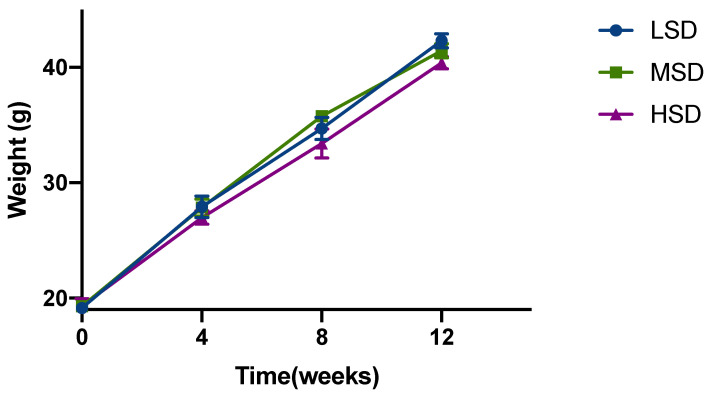
Growth performances of Kunming mice reared in different stocking density groups. Date presented as mean ± SD.

**Figure 2 animals-12-02538-f002:**
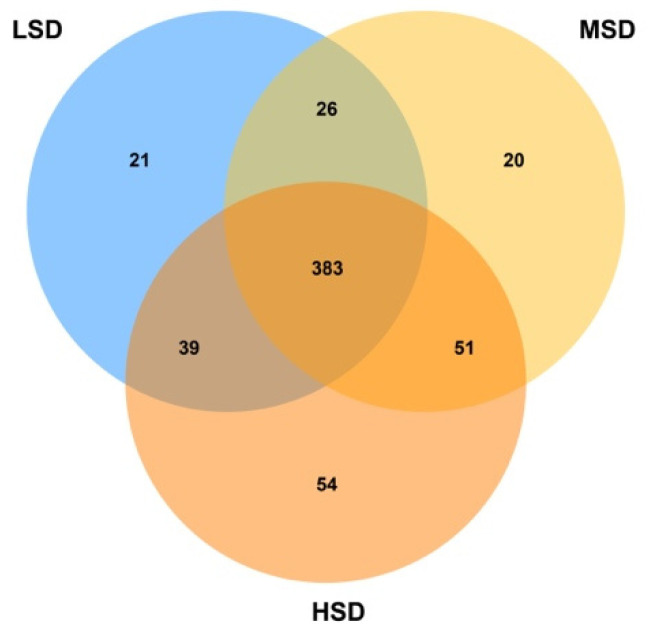
The intestinal contents of Kunming mice with different feeding densities were analyzed by 16S rRNA. Venn of bacterial communities (based on OTUs) in the sediment of treatments with different stocking densities in different density groups.

**Figure 3 animals-12-02538-f003:**
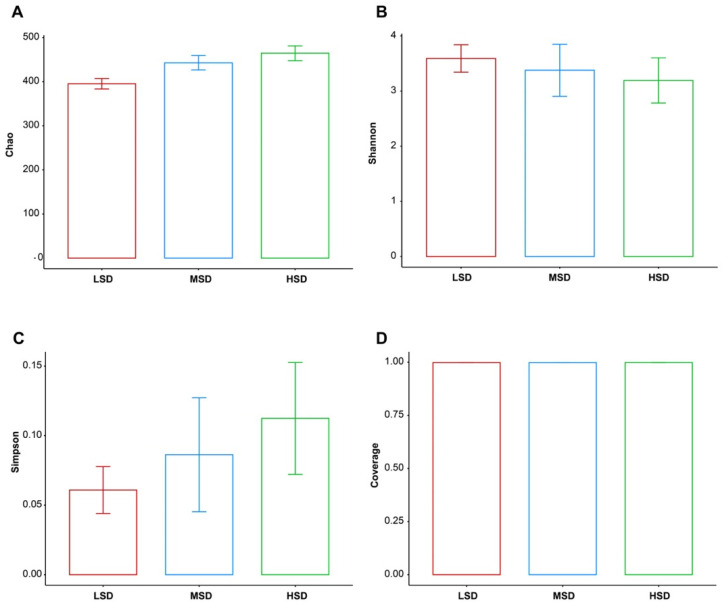
Diversity of the bacterial communities. (**A**) Chao estimator (Chao). (**B**) Shannon diversity index (Shannon). (**C**) Simpson diversity index (Simpson). (**D**) Coverage diversity index (Coverage). Date presented as mean ± SD.

**Figure 4 animals-12-02538-f004:**
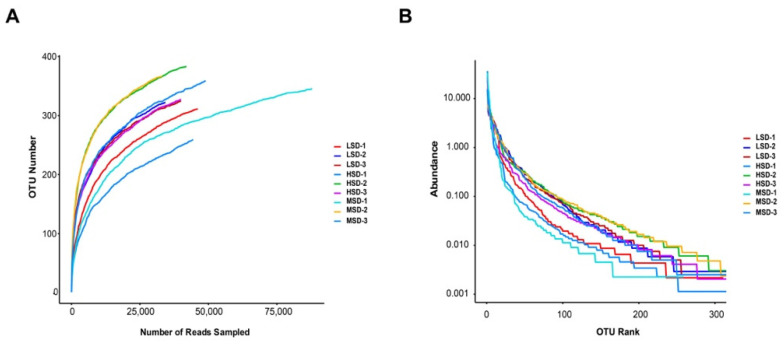
Analysis of gut microbial OTUs structures based on bacterial communities. (**A**) Rarefaction curves for sequencing of 16SrRNA in colorectal microflora. (**B**) Curves for diversity of samples by Rank-Abundance.

**Figure 5 animals-12-02538-f005:**
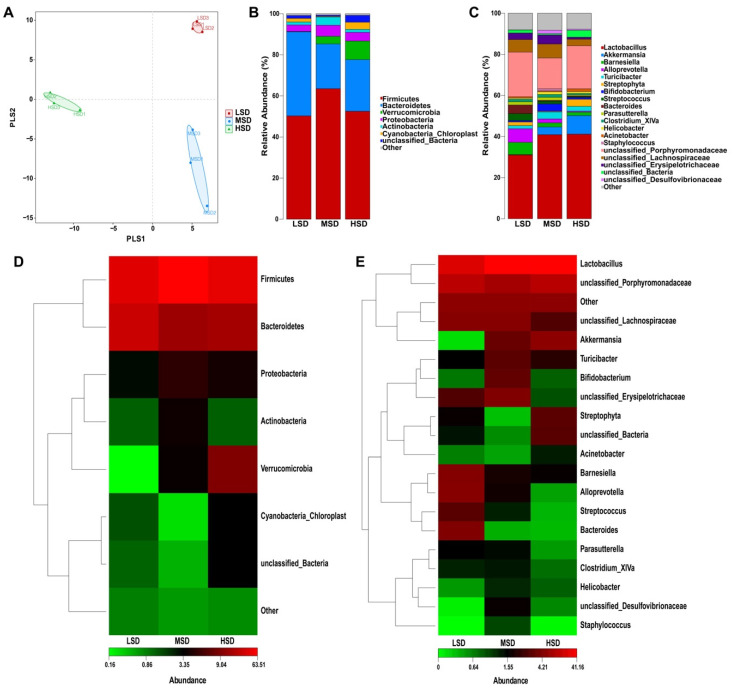
Multi-stage species composition analysis of single sample. (**A**) Partial least squares discriminant analysis (PLS-DA) of the samples. Each sample was represented as a point. (**B**) Relative abundance of the most abundant bacterial phylum. (**C**) Relative abundance of the most abundant bacterial genu. (**D**) The heat map showing the composition of the phylum-level microbiota combined with the results from the cluster analysis. (**E**) The heat map showing the composition of the genu-level microbiota combined with the results from the cluster analysis.

**Figure 6 animals-12-02538-f006:**
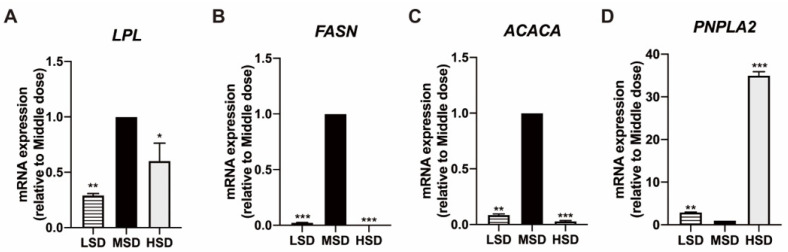
Effect of feeding density on fat metabolism in Kunming mice. (**A**) qPCR analysis was performed to assess the mRNA abundance of *LPL* in mice of different feeding density. (**B**) qPCR analysis was performed to assess the mRNA abundance of *FASN* in Kunming mice of different feeding density. (**C**) qPCR analysis was performed to assess the mRNA abundance of *ACACA* in mice of different feeding density. (**D**) qPCR analysis was performed to assess the mRNA abundance of *PNPLA2* in mice of different feeding density. Data were presented as a mean ± SD, n = 3, ** p* < 0.05, *** p* < 0.01, *** *p* < 0.001, vs. MSD group.

**Figure 7 animals-12-02538-f007:**
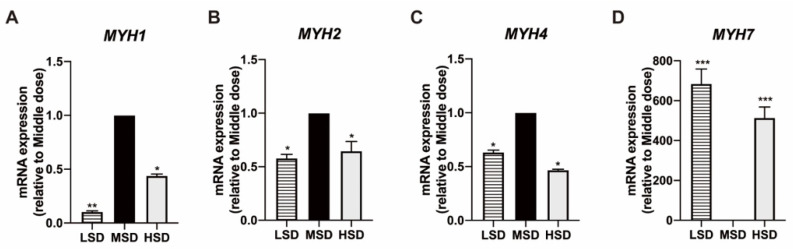
Effect of feeding density on muscle function of Kunming mice. (**A**) qPCR analysis was performed to assess the mRNA abundance of *MYH1* in mice of different feeding density. (**B**) qPCR analysis was performed to assess the mRNA abundance of *MYH2* in mice of different feeding density. (**C**) qPCR analysis was performed to assess the mRNA abundance of *MYH4* in mice of different feeding density. (**D**) qPCR analysis was performed to assess the mRNA abundance of *MYH7* in mice of different feeding density. Data were presented as a mean ± SD, n = 3, ** p* < 0.05, *** p* < 0.01, *** *p* < 0.001, vs. MSD group.

**Table 1 animals-12-02538-t001:** Primer Sequences.

Primer Name	Sequence (5’-3’)
*LPL*	TGGCGTAGCAGGAAGTCTGA TGCCTCCATTGGGATAAATGTC
*FASN*	GGCTCTATGGATTACCCAAGC CCAGTGTTCGTTCCTCGGA
*ACACA*	CGCCAACAATGGTATTGCAGC TCGGATTGCACGTTCATTTCG
*PNPLA2*	GGGTGCGCTATGTGGATGG CTCTCGCCTGAGAATGGGG
*MYH1*	CGGAGTCAGGTGAATACTCACG GAGCATGAGCTAAGGCACTCT
*MYH2*	TAAACGCAAGTGCCATTCCTG GGGTCCGGGTAATAAGCTGG
*MYH4*	AGGACCAACTGAGTGAAGTGA GGGAAAACTCGCCTGACTCTG
*MYH7*	AGACTGTCAACACTAAGAGGGT TGCCCCAAAATGGATTCGGAT
*GAPDH*	TGGCCTTCCGTGTTCCTAC GAGTTGCTGTTGAAGTCGCA

LPL, Lipoprotein lipase; FASN, Fatty acid synthase; ACACA, Acetyl-CoA carboxylases alpha; PNPLA2, Patatin Like Phospholipase Domain Containing 2; MYH1, Myosin heavy chain 1; MYH2, Myosin heavy chain 2; MYH4, Myosin heavy chain 4; MYH7, Myosin heavy chain 7; GAPDH, Glyceraldehyde-3-phosphate dehydrogenase.

**Table 2 animals-12-02538-t002:** Statistics of sequence data of all groups.

Sample Name	Raw Reads	Clean Reads	Effective Reads	Effective (%)
LSD	172,621	171,854	120,409	70.06
MSD	182,646	180,548	126,361	69.99
HSD	226,753	224,399	169,981	75.75

**Table 3 animals-12-02538-t003:** The bacterial abundance of Kunming mice fed with LSD, MSD, and HSD for 12 weeks in phylum.

Phylum	LSD (%)	MSD (%)	HSD (%)
Firmicutes	50.28	63.51	52.53
Bacteroidetes	40.93	21.78	25.14
Proteobacteria	3.15	5.35	4.24
Cyanobacteria_Chloroplas	1.75	0.32	3.46
Actinobacteria	1.46	4.01	1.46
unclassified_Bacteria	1.41	0.57	3.35
Candidatus_Saccharibacteria	0.84	0.66	0.76
Verrucomicrobia	0.16	3.74	9.04
Tenericutes	0.01246	0.00791	0.02118
Planctomycetes	0.00415	0.00712	0.00118
Ignavibacteriae	0.00166	0.00079	0.00118
Chloroflexi	0.00083	0.00079	0.00000
Deferribacteres	0.00083	0.03324	0.00235
Acidobacteria	0.00000	0.00000	0.00118

**Table 4 animals-12-02538-t004:** Statistics of bacterial abundance of Kunming mice in low density group, medium density group, and high stocking density group in genus.

LSD		MSD		HSD	
Genu	Percent (%)	Genu	Percent (%)	Genu	Percent (%)
Lactobacillus	31.05	Lactobacillus	40.91	Lactobacillus	41.16
unclassified_Porphyromonadaceae	21.73	unclassified_Porphyromonadaceae	14.99	unclassified_Porphyromonadaceae	20.96
Alloprevotella	6.58	unclassified_Lachnospiraceae	6.78	Akkermansia	9.04
unclassified_Lachnospiraceae	6.13	unclassified_Erysipelotrichaceae	4.46	Streptophyta	3.44
Barnesiella	5.99	Akkermansia	3.74	unclassified_Bacteria	3.35
Bacteroides	4.21	Bifidobacterium	3.58	unclassified_Lachnospiraceae	3.20
Streptococcus	3.37	Turicibacter	3.41	Turicibacter	2.36
unclassified_Erysipelotrichaceae	3.19	Barnesiella	2.02	Barnesiella	1.68
Streptophyta	1.73	Alloprevotella	1.90	Acinetobacter	1.35
Turicibacter	1.55	unclassified_Desulfovibrionaceae	1.70	unclassified_Erysipelotrichaceae	0.97

## Data Availability

The study’s original contributions are included in the article; further inquiries can be directed to the corresponding authors.

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
