# Peer review of "Effects of Stocking Density on Fatty Acid Metabolism by Skeletal Muscle in Mice"

_animals, 2022, doi:10.3390/ani12192538_

Round 1
Reviewer 1 Report
The manuscript, #animals-1855427, titled “Effects of Stocking Density on Intestinal Flora and Muscle Fatty Acids in Mice” by Qiuyan Chen et al. was to investigate the stocking density on dynamic distribution of gut microbiota, lipid metabolism and muscle mass of SPF grade Kunming mice. Their results showed that stocking density could regulate gut microbiota by changing the growth of intestinal bacteria, and reasonable stocking density could improve lipid metabolism and muscle quality to a certain extent. Due to the following concerns, I think it can be considered for major revisions.
General comments
1. Introduction: This study focuses on the effect of stocking density on basic guarantees for experimental animals. However, there were no related introduction about stocking density. The authors should introduce the background information of stocking density on experimental animals. This will help the readers understand the objective and significance of the study.
2. Line 77-85: (1) Please add the information that how many repetitions per treatment and how many mice per repetitions? (2) How does the authors determine the three stocking density treatments? What was the stocking density in the Material and Methods based on?
3. Line 101-112 and 114-118: (1) The description of 16S rRNA sequencing and bioinformatics analysis was confused. (2) The description of OTU was repeated in the above paragraphs.
4. Line 144-149: The statistics analysis should be doubled check. “Results between the two groups are compared using Student’s t test” will increase the probability of the two types of mistakes. The author should carry out statistics analysis by variance analysis.
5. Line 152-153, 241-242 and 288-289. Please double checked the data and added the statistics results. The author stated significant difference. However, we cannot find this statistic results from the showed figures and tables.
6. Discussion: Similarly, this study focuses on the effect of stocking density on basic guarantees for experimental animals. However, in the section of Discussion, the author discussed the relationship between gut microbiota and lipid metabolism and muscle mass of SPF grade Kunming mice. The authors should discuss the relationship between the stocking density and gut microbiota, lipid metabolism and muscle mass of mice.
Specific comments
1. Line 71: This statement should be added cited reference.
2. Line 101: Please revised “2.4.16. S rRNA sequencing and bioinformatics analysis” to “2.4. 16 S rRNA sequencing and bioinformatics analysis”
3. Line 139: Please revised “6s rRNA database” to “16S rRNA database”
4. Line 145: This sentence made the readers confused. Please revised.
5. Table 2: What does the meaning of “tag” in table 2? Did the authors mean “read”?
6. Line 188: This result was shown in figure 3D, not figure 4D.
7. Line 205 and Figure 5A: Please double-checked principal component analysis (PCA) or partial least squares discriminant analysis (PLS-DA) was used.
8. Line 328-329: Which results indicated that the mice in the MSD group had low mitochondrial content?
9. Please check the format of Reference. The title of this references should be just capitalized the first word.
Author Response
Point 1: Introduction: This study focuses on the effect of stocking density on basic guarantees for experimental animals. However, there were no related introduction about stocking density. The authors should introduce the background information of stocking density on experimental animals. This will help the readers understand the objective and significance of the study.
Response 1: Thanks for the reviewer’s suggestion, we have added the background information of stocking density on experimental animals in introduction.
Point 2: Line 77-85: (1) Please add the information that how many repetitions per treatment and how many mice per repetitions? (2) How does the authors determine the three stocking density treatments? What was the stocking density in the Material and Methods based on?
Response 2: Thanks for the reviewer’s suggestions, we have added relevant information in the materials and methods. According to the national standards of People's Republic of China, GB14925-2010 (Lavoratory animal-Requirements of environment and housing facilities), the minimum space for mice to live in is 1.196×10-3 cm3. The bottom area of our IVC cages is 4.4175×10-2 m2, high is 0.12 m, so we set the medium stocking density is 5 mice/cage, the low stocking density is 3 mice/cage, high stocking density is 10 mice/cage.
Point 3: Line 101-112 and 114-118: (1) The description of 16S rRNA sequencing and bioinformatics analysis was confused. (2) The description of OTU was repeated in the above paragraphs.
Resonse 3: We have modified the descriptions.
Point 4: Line 144-149: The statistics analysis should be doubled check. “Results between the two groups are compared using Student’s t test” will increase the probability of the two types of mistakes. The author should carry out statistics analysis by variance analysis.
Resonse 4: Thanks for the reviewer’s kindness, we have only analyzed the two groups difference, so we choice the Student’s t test.
Point 5: Line 152-153, 241-242 and 288-289. Please double checked the data and added the statistics results. The author stated significant difference. However, we cannot find this statistic results from the showed figures and tables.
Resonse 5: According to the reviewer’s suggestion, we have checked the data, and illustrated in the manuscript.
Point 6: Discussion: Similarly, this study focuses on the effect of stocking density on basic guarantees for experimental animals. However, in the section of Discussion, the author discussed the relationship between gut microbiota and lipid metabolism and muscle mass of SPF grade Kunming mice. The authors should discuss the relationship between the stocking density and gut microbiota, lipid metabolism and muscle mass of mice.
Resonse 6: Thanks for the reviewer’s advice, we have added the discussion of the relationship between the stocking density and gut microbiota, lipid metabolism and muscle mass of mice.
Point 7: Line 71: This statement should be added cited reference.
Resonse 7: We have added a reference here. Zhou, H.; Yu, B.; Sun, J.; Liu, Z.H.; Chen, H.; Ge, L.P.; Chen, D.W. Short-chain fatty acids can improve lipid and glucose metabolism independently of the pig gut microbiota. J Anim Sci Biotechno 2021, 12, doi:ARTN 61.
Point 8: Line 101: Please revised “2.4.16. S rRNA sequencing and bioinformatics analysis” to “2.4. 16 S rRNA sequencing and bioinformatics analysis”
Resonse 8: Thanks for the reviewer’s kindness, we are so sorry for this mistake, and we have revised the subtitle.
Point 9: Line 139: Please revised “6s rRNA database” to “16S rRNA database”
Resonse 9: Thanks for the reviewer’s kindness, we are so sorry for this mistake, and we have revised.
Point 10: Line 145: This sentence made the readers confused. Please revised.
Resonse 10: Thanks for the reviser’s advice, we have revised the sentence.
Point 11: Table 2: What does the meaning of “tag” in table 2? Did the authors mean “read”?
Resonse 11: Yes, we can also use “read”. Raw tags refer to the sequence of tags obtained by splicing double-ended readers, clean tags are the sequences of low quality and short length filtered by Raw tags, and Effective Tags are the sequences of tags that are finally used for subsequent analysis after filtering chimeras. Effective Ratio refers to the ratio of Effective tags to Raw Tags.
Point 12: Line 188: This result was shown in figure 3D, not figure 4D.
Resonse 12: Thanks for the reviewer’s kindness very much, we are so sorry for this mistake, and we have revised.
Point 13: Line 205 and Figure 5A: Please double-checked principal component analysis (PCA) or partial least squares discriminant analysis (PLS-DA) was used.
Resonse 13: Thanks for the reviewer’s kindness very much, we are so sorry for our mistake, we used PLS-DA there and we have revised in the manuscript.
Point 14: Line 328-329: Which results indicated that the mice in the MSD group had low mitochondrial content?
Resonse 14: This was inferred from others articles,but we didn’t verify, so we removed this sentence.
Point 15: Please check the format of Reference. The title of this references should be just capitalized the first word.
Resonse 15: We have checked the format of Reference. And the title of this references was right.

Reviewer 2 Report
Please find below comments to the manuscript entitled "Effects of Stocking Density on Intestinal Flora and Muscle Fatty Acids in Mice"
Line 2-the title requires modification as the following "Muscle Fatty Acids " is not accurate. I believe the author meant the fatty acid metabolism by skeletal muscle
line 16- 'mouse health, which may be related to gut microbiota'-the sentence requires revision
lines 17-18- sentence requires revision
lines 21-22- there is a big jump from gut microorganisms to lipid metabolism thus my suggestion is to provide an explanation how this correlates with one another
line 22- please change to 'lipid'.
line 28- what does it mean to have a 'reasonable gut microbiota'? I would suggest to replace reasonable with a different word
lines 27-31- it would be good to see the summary of the data obtained in addition to the text ; maybe a comparison of how LSD vs MSD and HSD vs MSD improved the parameters measured by the team
lines 31-32- the sentence does not add any new information. suggest to delete
lines 32-33- requires revision (language)
The abstract provides sufficient information about the study aim and general design however I would suggest to shorten the 'introductory' section and focus more on the actual results. I would like to know how many mice per cage were taken into the experiment and what were the main results (in numbers). The last sentences (29-33) are too general and does not encourage the reader to look into the actual data provided in the paper. Please revise the abstract and make it more concise and informative.
lines 37-50- requires substantial language revision
line 50- what did the author meant by 'muscle properties can be transmitted..'?
lines 56-57- please combine sentences
lines 71-74- requires substantial language revision
line 77- how many mice were used in the experiments?
line 87- 'forbidden to eat ' requires revision
line 126 - what the author meant by 'organization'? this requires amending
line 148- I would suggest to remove '**p<0.01 was considered extremely significant.'
Can you please explain why a negative correlation between the feeding and weight was observed after 12 weeks?
lines 154-155- how did the authors measure the 'general shape'
line 155- how did the authors measure the 'good mental state, sensitive response'?
line 157-how did the authors measure how much the mice were drinking the method was not described.
line 158- there is nothing mentioned about what the authors consider as normal/slow movement. How the movement was evaluated? the same is related to the 'lethargy and slow response. Please elaborate
lines 158-159- requires substantial language revision
Did the authors measure the weight of animals before the experiment? How often was the weight measured. It would be beneficial to show the %of body mass gain or loss as the current graph is not very informative; it shows the final weight of animals post experiment between groups.
Figure description requires revision.
Author Response
Point 1: Line 2-the title requires modification as the following "Muscle Fatty Acids " is not accurate. I believe the author meant the fatty acid metabolism by skeletal muscle
Response 1: Thanks for the reviewer’s kindness, we have changed "Muscle Fatty Acids " to fatty acid metabolism by skeletal muscle.
Point 2: line 16- 'mouse health, which may be related to gut microbiota'-the sentence requires revision
Response 2: Thanks for the reviewer’s kindness, we have revised this sentence.
Point 3: lines 17-18- sentence requires revision
Resonse 3: Thanks for the reviewer’s kindness, we have revised this sentence.
Point 4: lines 21-22- there is a big jump from gut microorganisms to lipid metabolism thus my suggestion is to provide an explanation how this correlates with one another
Resonse 4: Thanks for the reviewer’s advice, we have provided an explanation how gut microorganisms correlates with lipid metabolism.
Point 5: line 22- please change to 'lipid'.
Resonse 5: Thanks for the reviewer’s kindness, we have changed to ‘lipid’.
Point 6: line 28- what does it mean to have a 'reasonable gut microbiota'? I would suggest to replace reasonable with a different word
Resonse 6: Thanks for the reviewer’s kindness, we have replaced it with another explanation.
Point 7: lines 27-31- it would be good to see the summary of the data obtained in addition to the text ; maybe a comparison of how LSD vs MSD and HSD vs MSD improved the parameters measured by the team
Resonse 7: Thanks for the reviewer’s advice, we have modified there.
Point 8: lines 31-32- the sentence does not add any new information. suggest to delete
Resonse 8: Thanks for the reviewer’s suggestion, we have deleted the sentence.
Point 9: lines 32-33- requires revision (language)
Resonse 9: Thanks for the reviewer’s suggestion, we have revised here.
Point 10: The abstract provides sufficient information about the study aim and general design however I would suggest to shorten the 'introductory' section and focus more on the actual results. I would like to know how many mice per cage were taken into the experiment and what were the main results (in numbers). The last sentences (29-33) are too general and does not encourage the reader to look into the actual data provided in the paper. Please revise the abstract and make it more concise and informative.
Resonse 10: We have revised according to the reviewer’s suggestion.
Point 11: lines 37-50- requires substantial language revision
Resonse 11: Thanks for the reviewer’s kindness, we have revised here.
Point 12: line 50- what did the author meant by 'muscle properties can be transmitted..'?
Resonse 12: According to the reference [12], ‘muscle properties can be transmitted’ means skeletal muscle properties are transmissible via fecal microbiota transplantation.
Point 13: lines 56-57- please combine sentences
Resonse 13: Thanks for the reviewer’s kindness, we have combined the sentences.
Point 14: lines 71-74- requires substantial language revision
Resonse 14: Thanks for the reviewer’s suggestion, we have revised the language.
Point 15: line 77- how many mice were used in the experiments?
Resonse 15: Thanks for the reviewer’s advice, we have added the number of mice in the experiments.
Point 16: line 87- 'forbidden to eat ' requires revision
Resonse 16: Thanks for the reviewer’s suggestion, we have revised the short sentence.
Point 17: line 126 - what the author meant by 'organization'? this requires amending
Resonse 17: We have amended ‘organization’ to ‘tissue’.
Point 18: line 148- I would suggest to remove '**p<0.01 was considered extremely significant.'
Resonse 18: According to the reviewer’s suggestion, we have removed it.
Point 19: Can you please explain why a negative correlation between the feeding and weight was observed after 12 weeks?
Resonse 19: We have modified the result 3.1 in the manuscript.
Point 20: lines 154-155- how did the authors measure the 'general shape'
Resonse 20: In our original result, we visually observe these results. So, we deleted the related results in the manuscript.
Point 21: line 155- how did the authors measure the 'good mental state, sensitive response'?
Resonse 21: In our original result, we visually observe these results. So, we deleted the related results in the manuscript.
Point 22: line 157-how did the authors measure how much the mice were drinking the method was not described.
Resonse 22: We calculated the drinking water volume of each mouse based on the decrease volume in each cage every 24 hours.
Point 23: line 158- there is nothing mentioned about what the authors consider as normal/slow movement. How the movement was evaluated? the same is related to the 'lethargy and slow response. Please elaborate
Resonse 23: In our original result, we visually observe these results. So, we deleted the related results in the manuscript.
Point 24: lines 158-159- requires substantial language revision
Resonse 24: Thanks for the reviewer’s suggestion, we have edited the language.
Point 25: Did the authors measure the weight of animals before the experiment? How often was the weight measured. It would be beneficial to show the %of body mass gain or loss as the current graph is not very informative; it shows the final weight of animals post experiment between groups.
Resonse 25: Thanks for the reviewer’s advice, we supplied the weight of mice before the experiment in figure1. We added the bar according to the weight.
Point 26: Figure description requires revision.
Resonse 26: Thanks for the reviewer’s suggestion, we have revised figure description.

Reviewer 3 Report
The title is not in correlation with the content. You put on the title the expression ,,muscle fatty acids'' but you haven't quantifying the content of fatty acids from the mices' muscles. You analyzed the muscles function and the enzymes activity. You also have to change the name of the subchapter ,,3.5. Effects of stocking density on muscle fatty acide of SPF Kunming mice''.
The chapter ,,Introduction'' have to be specific and in deep correlation with your experiment. Please restructure the information you presented there. At the end of the chapter ,,Introduction'' add a phrase with the aim of the study.
Please mention the number of mices/cage and the cages dimensions, not only the stocking density.
Please mention the type of food the mice received during the experiment and mention if the food was the same for all 12 weeks or if it was changed.
Please mention the period of time for which the samples were stored in liquid nitrogen.
Please add at the chapter ,,Materials and Methods'' all the procedures that you used to obtain results. You haven't mentioned at the ,,Materials and Methods'' about the Chao index and Shannon index that you presented at the ,,Results''.
You, also, haven't mentioned about the weighting protocol. How you weight the mices?
All the abbreviations have to have the explanatory legend. Please add an explanatory legend for table 1, for the Primers names, so that even an unknowing reader to understand.
The ,,Conclusions'' contains insufficient information related to the experiments' results. Please add more information including some recommendations at the ,,Conclusions''.
Author Response
Point 1: The chapter ,,Introduction'' have to be specific and in deep correlation with your experiment. Please restructure the information you presented there. At the end of the chapter ,,Introduction'' add a phrase with the aim of the study.
Response 1: Thanks for the reviewer’s kindness, we have added the information according to the reviewer’s suggestions in the chapter ‘Introduction’.
Point 2: Please mention the number of mices/cage and the cages dimensions, not only the stocking density.
Response 2: According to the reviewer’s suggestions, we have added the number of mices/cage and cages dimensions in Materials and Methods.
Point 3: Please mention the type of food the mice received during the experiment and mention if the food was the same for all 12 weeks or if it was changed.
Resonse 3: We have added the type of food, and it wasn’t changed all 12 weeks according to the reviewer’s suggestions.
Point 4: Please mention the period of time for which the samples were stored in liquid nitrogen.
Resonse 4: Thanks for the review’s advice, we have added the information according to the reviewer’s suggestions.
Point 5: Please add at the chapter ,,Materials and Methods'' all the procedures that you used to obtain results. You haven't mentioned at the ,,Materials and Methods'' about the Chao index and Shannon index that you presented at the ,,Results''.
Resonse 5: We have added the information about the Chao index and Shannon index according to the reviewer’s suggestions at the ’Materials and Methods’.
Point 6: You, also, haven't mentioned about the weighting protocol. How you weight the mices?
All the abbreviations have to have the explanatory legend. Please add an explanatory legend for table 1, for the Primers names, so that even an unknowing reader to understand.
Resonse 6: We weighed each animal in the clean bench with a balance. Because each animal was ear tagged, body weight changes for each animal could be continuously recorded.
Point 7: All the abbreviations have to have the explanatory legend. Please add an explanatory legend for table 1, for the Primers names, so that even an unknowing reader to understand.
Resonse 7: Thanks for the reviewer’s suggestion, we have added the explanatory legend for table 1.
Point 8: The ,,Conclusions'' contains insufficient information related to the experiments' results. Please add more information including some recommendations at the ,,Conclusions''.
Resonse 8: Thanks for the reviewer’s advice, we have modified the conclusion.

Round 2
Reviewer 1 Report
No more comments.
Reviewer 3 Report
The authors corrected the manuscript according to reviewer recommendations.